# The Therapeutic Strategies Targeting Mitochondrial Metabolism in Cardiovascular Disease

**DOI:** 10.3390/pharmaceutics14122760

**Published:** 2022-12-09

**Authors:** Xiaoyang Huang, Zhenhua Zeng, Siqi Li, Yufei Xie, Xiaoyong Tong

**Affiliations:** 1Department of Pharmacology and Pharmacy, School of Pharmaceutical Sciences, Chongqing University, Chongqing 401331, China; 2Biomedical Research Center, Hunan University of Medicine, Huaihua 418000, China; 3Central Clinical School, Monash University, Melbourne, VIC 3004, Australia; 4Jinfeng Laboratory, Chongqing 401329, China

**Keywords:** cardiovascular disease, mitochondrial metabolism, mitochondrial calcium, mitochondrial dynamics, reactive oxygen species, mitochondrial DNA, mitochondria-targeted therapy, gene therapy

## Abstract

Cardiovascular disease (CVD) is a group of systemic disorders threatening human health with complex pathogenesis, among which mitochondrial energy metabolism reprogramming has a critical role. Mitochondria are cell organelles that fuel the energy essential for biochemical reactions and maintain normal physiological functions of the body. Mitochondrial metabolic disorders are extensively involved in the progression of CVD, especially for energy-demanding organs such as the heart. Therefore, elucidating the role of mitochondrial metabolism in the progression of CVD is of great significance to further understand the pathogenesis of CVD and explore preventive and therapeutic methods. In this review, we discuss the major factors of mitochondrial metabolism and their potential roles in the prevention and treatment of CVD. The current application of mitochondria-targeted therapeutic agents in the treatment of CVD and advances in mitochondria-targeted gene therapy technologies are also overviewed.

## 1. Introduction

Cardiovascular disease (CVD) remains the leading cause of death and disease burden worldwide [1]. Energy-demanding organs such as the heart are affected by mitochondrial function, and the relationship between mitochondrial metabolism and CVD has been widely proved [2,3,4,5,6,7]. Mitochondrial metabolism is a highly complex energy-releasing process involving a series of enzymatic reactions, during which sugars, fats, and proteins are oxidized through the tricarboxylic acid cycle (TCA) and oxidative phosphorylation (OXPHOS). Mitochondrial dysfunction is a symbol of the exacerbated intracellular environment leading to diseases such as CVD [2,3,4,5,6,7]. Regulating mitochondrial metabolism is a promising therapeutic strategy for CVD. This review focuses on the therapeutic strategies of mitochondrial metabolism in CVD in hope of summarizing the advances in this field.

## 2. Mitochondrial Metabolism Dysfunction in CVD

Mitochondrial metabolism is the main source of energy supply for most cells, especially for energy-consuming cells, such as cardiomyocytes. The glucose OXPHOS in the mitochondria is the major metabolism pathway in most cells. OXPHOS is, indeed, a more efficient means of generating ATP than glycolysis [8]. Mitochondrial metabolic dysfunction has been widely reported in CVD, characterized by metabolic reprogramming [9]. In pulmonary arterial hypertension (PAH), a significant metabolic reprogramming is observed, that is, in pulmonary artery smooth muscle cells (PASMCs), the level of OXPHOS is significantly reduced, and the levels of glycolysis and fatty acid oxidation (FAO) are significantly increased [10,11]. Similar metabolic reprogramming from glucose OXPHOS to glycolysis occurs in aortic smooth muscle cells (SMCs) in the development of aneurysms and atherosclerotic diseases [5,12]. The major metabolism pathway in cardiomyocytes is FAO. During the progression of heart failure, metabolism shifts from majorly relying on FAO to OXPHOS in the early stage then leans to glycolysis in the late stage [13]. These studies indicate that mitochondrial metabolic disorders may be the important driving force in the development of CVD. Therefore, exploring the factors affecting mitochondrial metabolism and the molecular mechanisms of mitochondrial metabolic reprogramming in CVD have important theoretical implications for the development of mitochondria-targeted therapeutics for CVD by regulating metabolism.

## 3. Major Factors in Mitochondrial Metabolism

### 3.1. Mitochondrial DNA

Mitochondria are the only semi-autonomous genetic organelles in mammalian cells with their own transcriptional and translational system independent of the nucleus. Mitochondria autonomously synthesize a variety of key proteins and enzymes by mitochondrial DNA (mtDNA). mtDNA is mainly distributed in the mitochondrial matrix and inner membrane and contains a double-stranded circular DNA with a molecular weight of 16.5 kb, 16,569 nucleotide pairs, coding sequences of 37 genes, 2 ribosomal RNAs, 22 transporter RNAs, and 13 genes encoding important component proteins that are closely related to mitochondrial OXPHOS [14,15]. mtDNA plays an extremely important role in providing the template for translating enzymes that maintain the normal mitochondrial OXPHOS process and cell function. The enzyme includes 7 mitochondrial oxidative respiratory chain NADH dehydrogenase subunits, 1 cytochrome B, 3 cytochrome C oxidase subunits, and 2 ATP synthetases [16]. Due to the absence of introns and histones in mtDNA, and the lack of a similar DNA repair system in the nucleus, mtDNA is more susceptible to oxidative damage by reactive oxygen species (ROS) and reactive nitrogen species (RNS) [17]. Furthermore, DNA mutations in the organism are more likely to occur in mtDNA, which may be related to the high error rate of mtDNA replication [18]. Therefore, some researchers assert that mtDNA is an outpost of cytogenetic stress [19]. mtDNA damage can reduce the activity of the mitochondrial oxidative respiratory complex, and inhibit OXPHOS, leading to depressed vasodilatory function [20] and metabolic heart disease accompanied by PAH [21].

The number of mitochondria in different tissues and cells varies based on energy requirements and metabolic demand, as well as the copy number of mtDNA. Therefore, the quantification of mtDNA copy number is used to evaluate mitochondrial function. Large population studies on CVD find that the population with reduced mtDNA copy number is more susceptible to suffering from diabetic metabolic syndrome [22], heart failure [23], peripheral arterial disease-related all-cause mortality and cardiovascular events [24], and other CVD [25]. Thus, mtDNA stability is an important internal factor affecting mitochondrial metabolism and metabolic-related CVD.

### 3.2. Mitochondrial Dynamics

Mitochondrial dynamics refers to the dynamic cycle of biogenesis, fusion, fission, and degradation that mitochondria continuously undergo to maintain their integrity, distribution, and morphology [26]. These processes reflect the early adaptation of mitochondria in response to external stimuli, maintaining overall mitochondrial function by removing irreversibly damaged parts and preserving the functional parts. In different tissues and cells, mitochondria morphology and network structure vary. In cardiomyocytes, mitochondria are numerous, covered with ridges, and spherical in shape, while in lymphocytes, the mitochondria are few in the count, tube-shaped. This also shows that the correlation of different morphological features of mitochondria with the energy requirements of tissue cells. Increasing evidences show that the disturbance of mitochondrial dynamics is extensively involved in the progression of CVD, including heart failure [27,28], myocardial ischemia/reperfusion (I/R) injury [29], abdominal aortic aneurysm (AAA) [30] and PAH [31,32].

#### 3.2.1. Mitochondrial Fusion

Mitochondrial fusion is an important process that enables the exchange of DNA, proteins, lipids, and metabolites between different mitochondria [33]. Mitochondrial fusion consists of two processes, outer and inner membrane fusion, regulated by the nuclear-encoded dynamin-related GTPases protein family membrane fusion proteins 1 (Mfn1), membrane fusion proteins 2 (Mfn2), and optic atrophy protein 1 (OPA1). The outer membrane fusion is dependent on Mfn1 and Mfn2, while OPA1, located in the inner mitochondrial membrane, is involved in the inner mitochondrial membrane fusion. Numerous studies have shown that mitochondrial fusion is involved in mitochondrial metabolism [34]. Mfn2 can interact with pyruvate kinase isozyme type M2 (PKM2) to promote mitochondrial fusion, facilitate OXPHOS, and inhibit glycolysis [35]. In both yeast and mammalian cells, blocking mitochondrial fusion can lead to mitochondrial dysfunction and inadequate ATP supply [36]. Loss-of-function mutations of Mfn1 and Mfn2 cause: (1) decreased cellular glucose oxidation level, (2) depolarization of the mitochondrial membrane potential, (3) inhibition of the mitochondrial TCA and oxidative respiration, (4) conversion of cellular energy metabolism from OXPHOS to glycolysis, and (5) severe cellular functional defects. Whereas stress-induced mitochondrial fusion restores mitochondrial ATP production [37]. When mitochondrial OXPHOS is blocked, it causes an increase in mitochondrial ROS (mtROS), which triggers oxidative damage to mitochondria and increases mtDNA mutations. However, mitochondrial fusion can replenish normal DNA to some mitochondria that contain missing or damaged DNA to “repair” mtDNA and ultimately maintain mitochondrial function normally. It has been confirmed by crossing two Hela cell lines with mitochondrial OXPHOS dysfunction caused by mutations at different sites of mtDNA, researchers find that the OXPHOS deficient cells are repaired by mitochondrial fusion [38]. Mitochondrial fusion can maintain normal mitochondrial metabolism function by “repairing” damaged mtDNA through sharing normal mtDNA [39], and inhibition of mitochondrial fusion can cause loss of mtDNA [40]. Besides, promoting mitochondrial fusion is beneficial to cardiovascular protection [26].

#### 3.2.2. Mitochondrial Fission

Mitochondrial fission is mainly mediated by mitochondrial fission protein 1 (Fis1) and dynamin-related protein 1 (Drp1) in the cytoplasm. Under physiological conditions, mitochondrial division can serve the purpose of isolating irreversibly damaged DNA or toxic material from the mitochondria, and defective mitochondria are removed via the mitophagy pathway when the mitochondrial membrane potential and pH gradient fail to meet the requirements for fusion. Thus, moderate mitochondrial division is beneficial to maintain normal cellular energy metabolism. However, excessive mitochondrial fission inhibits the respiratory chain, represses ATP production, and leads to cellular dysfunction [41,42]. A study on the changes in ATP production in the circadian rhythm of fibroblasts finds that the circadian rhythm of ATP production is closely related to mitochondrial morphology and that increased mitochondrial fission causes uneven transcriptional production of oxidative respiratory complexes I, IV and V and ATP [43], which suggest that maintaining the balance of mitochondrial fission and fusion is essential for maintaining mitochondrial metabolism, furthermore excessive mitochondrial fission may lead to metabolism-related diseases, such as diabetic cardiomyopathy [44] and heart failure [45,46,47].

### 3.3. Mitochondrial Calcium

#### 3.3.1. Mitochondrial Calcium Transport

Dynamic localization of intracellular organelles with calcium storage functions such as sarcoplasmic/endoplasmic reticulum (SR/ER) and mitochondria allows the formation of high Ca^2+^ content microdomains formed between ER and mitochondria named mitochondrial associated membranes (MAMs) [48]. As a second messenger, Ca^2+^ is widely involved in cellular signaling and physiological activities. Under resting conditions of the cell, the cytoplasmic Ca^2+^ ([Ca^2+^]_c_) concentration is much lower than that of mitochondria, and the Ca^2+^ uptake by mitochondria is strictly restricted. With stimulation, the concentration of [Ca^2+^]_c_ surpasses the threshold, and mitochondria rapidly uptake Ca^2+^ to counteract the excessive [Ca^2+^]_c_.

There are special channels for mitochondria to transport Ca^2+^ where Ca^2+^ can shuttle through the cytosolic and mitochondrial membrane gap via the calcium-permeable voltage-dependent anion channels (VDACs). VDACs are one of the most abundantly expressed proteins in the outer mitochondrial membrane and contain three isoforms, VDAC1, VDAC2, and VDAC3 [49]. VDAC isoforms are resembling in structure and function, but the three isoforms differ in Ca^2+^ affinity, tissue expression specificity, and the regulation of mitochondrial biological processes. Apart from Ca^2+^, VDACs also allow the transportation of the electron transport chain (ETC) substrates pyruvate, succinate, and NADH as well as the dephosphorylation between ATP and ADP. VDACs mainly locate in MAMs and ER forming a complex with inositol 1,4,5-trisphosphate receptor (IP3R), and together with adapter protein GRP75 in the ER, forming the ER-mitochondria high Ca^2+^ microdomains to facilitate ER calcium delivery to mitochondria [50]. Knockdown of VDACs disrupts IP3R-mediated mitochondrial Ca^2+^ ([Ca^2+^]_m_) transport, and overexpression of either isoform of VDACs restores [Ca^2+^]_m_ uptake [50]. Ca^2+^ transport by the inner mitochondrial membrane is strictly controlled by the mitochondrial calcium uniporter (MCU), which is highly selective for Ca^2+^. The MCU pore contains [Ca^2+^]_m_ uptake proteins MICU1, MICU2, and MICU3, the MCU basic regulator, EMRE, and the major negative subunits of MCU, MCUb, and the MCU regulator, MCUR1 [51]. The activity of MCU is regulated by both [Ca^2+^]_c_ and [Ca^2+^]_m_ [52]. To maintain the [Ca^2+^]_m_ homeostasis, [Ca^2+^]_m_ can be excreted from mitochondria through efflux channels. Another Ca^2+^ pump, Na^+^/Ca^2+^ exchanger (NCX) locates in both the cell membrane and inner mitochondrial membrane for maintenance of [Ca^2+^]_m_ homeostasis [53]. NCX allows the exchange of three Na^+^ with one Ca^2+^ across the membrane at the same time, to expel [Ca^2+^]_m_ from the matrix [54]. Under physiological conditions, [Ca^2+^]_c_ and [Ca^2+^]_m_ concentrations stay stable to sustain normal energy requirements. However, exogenous stimuli or pathological conditions may cause imbalanced [Ca^2+^]_m_ homeostasis, causing decreased mitochondrial ATP production and metabolism-related diseases. Whereas all these systems allow Ca^2+^ to enter and efflux, the mitochondria constitute essential tools for the maintenance of [Ca^2+^] homeostasis, and any perturbation may cause [Ca^2+^]_m_ imbalance and associated disease.

#### 3.3.2. [Ca^2+^]_m_ and Mitochondrial Metabolism

A growing number of studies have shown that an imbalance in [Ca^2+^]_m_ homeostasis is an important cause of CVD [55]. For example, heart-specific knockdown of VDAC2 has been found to cause [Ca^2+^]_m_ imbalance and dilated cardiomyopathy [56]. Similarly, a lack of MCU has been found to promote PASMC proliferation and apoptosis resistance in human and experimental PAH models, and restoring MCU expression can alleviate the symptoms of experimental PAH [57]. Therefore, the maintenance of [Ca^2+^]_m_ homeostasis is essential to maintaining the normal energy metabolism of mitochondria and the function of the body. [Ca^2+^]_m_ imbalance can be caused by [Ca^2+^]_m_ overload. Numerous studies have shown that when [Ca^2+^]_m_ is abnormally increased, may cause [Ca^2+^]_m_ overload, resulting in oxidative stress, mitochondrial structural damage, and dysfunction of mitochondria, which in turn reduces ATP production and triggers cell apoptosis [2,58,59,60,61]. Nevertheless, [Ca^2+^]_m_ itself is a key factor in regulating mitochondrial ATP production by participating in the following processes. Ca^2+^ can activate pyruvate dehydrogenase complex (PDC), a complex enzyme composed of multiple subunits [62], isocitrate dehydrogenase (IDH), a key TCA rate-limiting enzyme [63], and 2-oxoglutarate dehydrogenase complex (OGDC) in the mitochondrial matrix [64,65,66]. When glucose is converted to pyruvate in the cytosol, it is transported to the mitochondria and catalyzed by PDC to acetyl coenzyme A (acetyl-CoA). Acetyl-CoA undergoes the TCA producing carbon dioxide, GTP, and NADH. Ca^2+^ can enhance PDC activity by binding to pyruvate dehydrogenase (PDH) phosphatase subunit 1 (PDP1), which accelerates the conversion of pyruvate to acetyl-CoA [67,68] and promote the ATP production through dephosphorylation of the PDH subunit [69]. Meanwhile, Ca^2+^ can accelerate the TCA by enhancing the activity of IDH [70] and OGDC [71]. In addition, the activation of these dehydrogenases by Ca^2+^ increases the production of NADH, which provides electrons to the ETC and enhances the production of ATP [72]. However, when [Ca^2+^]_m_ overload happens, the production of ATP and other physical processes are hindered. [Ca^2+^]_m_ overload could result from a stress-induced sustained increase of [Ca^2+^]_c_ or ryanodine receptor 2 mediated ER Ca^2+^ leak [73,74,75]. [Ca^2+^]_m_ overload causes an increased positive charge of the inner mitochondrial membrane, leading to depolarization of the mitochondrial membrane, decreasing the mitochondrial membrane potential, and impeding electron transport. [Ca^2+^]_m_ overload promotes the production of mtROS and inhibits mitochondrial oxidative respiration by impairing the activity of complex enzymes on the ETC, causing downregulation of ATP production [2,76]. In addition, [Ca^2+^]_m_ overload stimulates the opening of mitochondrial membrane permeability transition pores (MPTP), causing the release of cytochrome C and leading to apoptosis [77,78].

### 3.4. ROS

ROS refer to a class of free radicals composed of oxygen molecules with an odd number of electrons, characterized by a short half-life and high activity. Mitochondria are the main source of ROS when electrons leak during transfer and are captured by oxygen molecules. During OXPHOS, mitochondria transfer electrons from electron donors to electron acceptors through the electron transfer chain, catalyze the generation of ATP and reduce oxygen molecules to water. Some studies have speculated that about 0.2–2% of these consumed oxygen molecules are converted to ROS [79]. In physiological conditions, low levels of ROS can act as signaling molecules that participate in mitochondrial metabolism [80,81,82,83,84], and promote adaptive upregulation of antioxidant enzymes to maintain the body’s health [85]. ROS could be rapidly scavenged by antioxidant enzymes, such as superoxide dismutase 2 (SOD2), glutathione reductase, and catalase. When ROS exceed the clearance capacity of the antioxidant system, excessive ROS may damage proteins, lipids, and mtDNA in mitochondria causing oxidative stress and diseases [86], such as CVD [87]. Therefore, to avoid cellular dysfunction, redox homeostasis must be strictly controlled in case of excessive ROS accumulation [88]. Studies have shown that excessive ROS can directly inactivate aconitase by interacting with its iron-sulfur clusters [89]. In TCA, H_2_O_2_ can react directly with α-ketoacids, pyruvate, oxaloacetate, and 2-oxoglutarate and then cause mitochondrial metabolic dysfunction [90]. Some studies have used the reaction of pyruvate and H_2_O_2_ to consume H_2_O_2_ which prevents cells from oxidative stress damage and inhibits the opening of MPTP caused by oxidative stress, reversing myocardial ischemia [91]. ROS can also cause mtDNA damage [92,93], promote mitochondrial division [94,95], and induce mitochondrial autophagy [96], leading to mitochondrial metabolic dysfunction. To sum up, mitochondria-derived ROS can cause great redox stress to mitochondria that leads to mitochondrial metabolic dysfunction and CVD. The major factors and pathways involved in mitochondrial metabolism in the physiological and pathological conditions of the cardiovascular system are illustrated in Figure 1.

## 4. Mitochondrial Metabolism Disorder and CVD

### 4.1. PAH

PAH is a heterogeneous and fatal disease of the pulmonary vasculature, clinically defined as resting mean pulmonary artery pressure >20 mmHg, and normal left atrial pressure and pulmonary vascular resistance ≥3 Wood units [97]. PAH is characterized by occlusive remodeling and increased resistance caused by persistent pulmonary vasoconstriction, leading to right ventricular hypertrophy and heart failure. Excessive proliferation and apoptosis resistance of PASMCs and pulmonary artery endothelial cells (PAECs) are the main causes of pulmonary vascular remodeling.

The association of metabolic reprogramming and PAH has become a consensus over the past decades [98]. In PAH, a shift of energy metabolism from OXPHOS of glucose to cytoplasmic glycolysis in PASMCs and PAECs has been widely demonstrated in human lung-derived cell cultures and multiple mouse models of PAH, and the prevailing view is that the driving force for this metabolic shift stems from the pathological accumulation of hypoxia-inducible factor 1α (HIF-1α) [99]. HIF-1α is a hypoxia-inducible protein that significantly promotes glycolytic metabolism and is a core regulator of the body’s adaptive repair of the intracellular oxygen environment, playing an important role in a variety of physiological functions including cell proliferation, pro-survival, angiogenesis and metabolism. In the development of PAH, HIF-1α accumulation is accompanied by an increase in glycolysis and glycolysis-related enzyme activities [10]. In contrast, mitochondrial glucose OXPHOS decreases significantly in the pulmonary artery. However, it is noteworthy that the levels of enzymes related to fatty acid metabolism and FAO in mitochondria of PASMCs and PAECs are significantly increased in a variety of PAH models [100,101,102]. Carnitine palmitoyl transferase 1 (CPT1) locates in the outer mitochondrial membrane to convert acyl-CoA species to their corresponding acyl-carnitines for transporting into the mitochondria. The upregulation of CPT1 in PASMCs increases ATP production and promotes PASMCs proliferation, while inhibition of CPT1 blocks the proliferation of PASMCs [11]. Knockdown of malonyl-CoA decarboxylase, an enzyme related to FAO, inhibits monocrotaline (MCT) or hypoxia-induced PAH in mice [103], suggesting that the increased FAO in PAH may be a compensatory pathway for the adaptation of pulmonary vascular cells to energy defects, but it may also be one of the reasons for the proliferation of PASMCs. Unlike the pulmonary vasculature, in PAH, fatty acid accumulation is increased in the right ventricular tissue, but the level of FAO is significantly decreased and glycolysis is enhanced, presumably related to the fact that cardiac tissue itself derives its energy from FAO [104,105,106]. In addition, numerous studies have shown that mitochondrial metabolism disorders caused by an imbalance of the redox system, [Ca^2+^]_m_ and excessive mitochondrial fission can promote the proliferation of PASMCs, leading to pulmonary vascular remodeling [57,107,108,109], while improving mitochondrial metabolism can alleviate PAH [109,110,111]. These studies indicate that mitochondrial metabolism disorder is one of the important driving factors in PAH, and improving mitochondrial metabolism may be a potential target for the treatment of PAH.

### 4.2. Aortic Aneurysm

Aortic aneurysm is an aortic dilatation disease caused by multiple factors, characterized by irreversible dilatation of the aortic diameter beyond 50% of the normal diameter or the abdominal aortic diameter ≥3 cm. Risk factors for aneurysms include smoking, age, and genetics. When an aortic aneurysm occurs, the risk of aneurysm rupture gradually increases parallel with the continued expansion of the aorta and the enlargement of the aneurysm. Once the aneurysm ruptures, the chance of fatality will be greater than 80% [112]. Moreover, due to the insidious nature of aneurysm development, patients may not show any clinical symptoms before the aneurysm rupture. Therefore, aneurysm rupture is the greatest risk factor for death in patients with an aneurysm. Because of the lack of a specific drug and the short time from onset to death, early screening is an important clinical precaution method, for aneurysms larger than 5.5 cm in diameter, the main treatment is endovascular aortic repair and surgery. It is urgent to determine the pathogenesis of aneurysms and develop targeted drugs. The main features of aneurysms are excessive proliferation and migration of aortic SMCs, increased secretion of extracellular matrix and degradation of elastic fiber, which cause vascular remodeling. Inflammation and redox imbalance are considered important drivers of aneurysm development. Chronic vascular inflammation can recruit immune cells, amplify the inflammatory response, cause SMCs dysfunction and apoptosis, and eventually lead to aortic smooth muscle tissue damage, causing an aneurysm [113].

In recent years, the pathological role of mitochondrial metabolism in aneurysms has attracted attention [114,115]. Single-cell transcriptomic analysis of ascending aortic tissues from eight patients with ascending thoracic aortic aneurysm (ATAA) and three healthy controls finds that, compared with healthy controls, the number of immune cells (especially T lymphocytes) in patients with ATAA increases significantly, while the number of non-immune cells is decreased, indicating the presence of inflammatory response in patients with aortic aneurysm. The differential gene expression data analysis suggests extensive mitochondrial dysfunction in ATAA patients [116]. In addition, transcriptomic and metabolomic analyses of the aorta in patients with Marfan syndrome (MFS, the major complication of which is the risk of ATAA) and mouse models reveal that the decreased expression of mitochondrial transcription factor A (TFAM) and mtDNA content in the aorta of patients and model groups repress the expression of mitochondrial oxidative respiration complex and mitochondrial function related proteins, instead, enhance the glycolytic related proteins expression, and eventually decrease the cellular oxygen consumption rate and increase the production of lactate. However, administration of nicotinamide riboside, a precursor of NAD, restores the mitochondrial metabolism and rapidly reverses the aneurysm in mice, indicating that mitochondrial dysfunction and mtDNA depletion are a new hallmark of MFS aneurysm disease [117]. Similarly, in a study on the relationship between mitochondrial respiration and fatal aortic dissection in patients and in angiotensin II-induced aortic aneurysms in ApoE knockout mice, the mitochondrial respiration-related proteins and the cell oxygen consumption rate are reduced, but the production of lactate is increased, while nicotinamide riboside treatment can reduce the development of aortic aneurysms and sudden death caused by aortic aneurysm rupture in mice [5]. These findings suggest that mitochondrial metabolism is involved in the development of aneurysms, and targeting mitochondrial metabolism may be a novel clinical strategy for the prevention and treatment of aneurysms.

### 4.3. Atherosclerosis

Atherosclerosis, a vascular disease characterized by lipid deposition and inflammation in the arterial wall, is the pathological cause of coronary heart disease and peripheral vascular diseases imposing a serious risk to health. Vascular endothelial cells (ECs), macrophages, and SMCs are the main cell types chronologically involved in the development of atherosclerosis. As initiates with hypercholesterolemia-induced endothelial dysfunction [118], damaged ECs trigger inflammation to recruit monocytes. Monocytes are induced to M1 macrophages, which accelerate the local inflammatory reaction and promote abnormal proliferation of SMCs leading to formation of atherosclerosis. Research has shown that abnormal mitochondrial metabolism in macrophages [119] and SMCs [12] is closely related to the development of atherosclerosis.

Glycolysis is the main form of energy metabolism in ECs and M1 macrophages [120,121]. Studies have found that the expression of glycolysis-related enzymes and glycolysis in ECs are significantly down-regulated in atherosclerosis [122]. In contrast, in atherosclerosis, inflammatory factors can promote the metabolic reprogramming of M1 macrophages from myelocytomatosis viral oncogene-dependent to HIF-1α-dependent, which enhances glycolysis [123]. While under hypoxia, either knockdown of HIF-1α or inhibition of the expression of glycolytic related protein 6-phosphofructo-2-kinase/fructose-2,6-biphosphatase 3 (PFKFB3) can significantly hamper the glycolysis in macrophages, reduce the expression of tumor necrosis factor-alpha (TNF-α), and promote macrophage apoptosis [124]. Similarly, in M1 macrophages, inhibition of miR-33 inhibits glycolysis, promotes the expression of aldehyde dehydrogenase (ALDH) family 1, subfamily A2 (ALDH1A2), and increases OXPHOS. Inhibition of miR-33 in vivo could also change the phenotype of macrophages in the plaque by activating M2 macrophages and inhibiting the progression of high-fat diet-induced atherosclerosis [125]. Different from M1 macrophages, M2 macrophage’s main supply energy is through FAO [121] and exhibit anti-inflammatory and anti-atherogenic properties [126]. The major energy metabolism of aortic SMCs is through glucose OXPHOS. It has been found that in aortic SMCs near the plaque region in atherosclerotic mice, energy metabolism of SMCs switches from glucose OXPHOS to glycolysis along with the upregulation of glucose transporter 1 (Glut1), glucose uptake and the glycolysis. Overexpression of Glut1 in SMCs in a metabolic syndrome mouse model can promote the development of atherosclerosis [12]. These findings show that the abnormal energy metabolism in the major players is associated with development of atherosclerosis. Further mechanism of mitochondrial metabolism involved in atherosclerosis deserves exploration.

### 4.4. Heart Failure

Cardiomyocytes contains large numbers of mitochondria as heart is one of the most energy-consuming organs of the body. Thus, disturbances in mitochondrial metabolism can lead to an inadequate energy supply to the heart and cause cardiac dysfunction. The reprogramming of cardiac energy metabolism during the development of heart failure has been widely demonstrated [13,127,128]. Under physiological conditions, about 40–60% of the energy required for myocardial activity derives from mitochondrial FAO, and the rest mainly comes from the oxidation of sugars, ketones and amino acids [129]. It has been reported that in the early stage of heart failure, the heart maintains normal FAO through a compensatory way, but the increased volume and pressure load caused by sustained sympathetic excitation can cause cardiac hypertrophy. However, during the compensatory stage, proteins responsible for fatty acid transport, such as CPT1, is significantly down-regulated, leading to a significant decrease in FAO. Meanwhile, cardiomyocytes compensate for the lack of myocardial energy by increasing glucose oxidation [13], which is also known as the “embryonic” re-evolution of cardiac energy metabolism. The influence on FAO during the compensatory phase is inconsistent in different models of heart failure [129]. When severe heart failure happens, blood and oxygen deficiency can trigger insulin resistance in cardiomyocytes, resulting in reduced aerobic oxidation of glucose, a shift in energy metabolism to glycolysis, and the generation of lactate to further aggravate heart failure, which end up in decompensated phase. Therefore, the shift of mitochondrial energy metabolism is one of the important characteristics of heart failure. During which, increased mtROS [130], disturbed calcium homeostasis [131], abnormal mitochondrial quality control (such as decreased mitochondrial biogenesis [27], increased mitochondrial fission [132] and suppressed mitophagy, etc.) may also be important reasons for the reprogramming of myocardial mitochondrial energy metabolism, and study also has shown that these factors are extensively involved in the progression of heart failure [133]. Therefore, targeted regulation of cardiomyocyte energy metabolism and mitochondrial dysfunction has also become an important strategy in the treatment of heart failure [129,133]. For example, Timothy Wai et al. finds that in yeast mitochondrial escape protein 1L knockout mice, the upregulation of Oma1 under stress leads to the cleavage of L-OPA1 and myocardial mitochondrial fragmentation, induction of shift in myocardial mitochondrial energy metabolism from FAO to glucose metabolism, ultimately causing myocardial cell necrosis, fibrosis and ventricular remodeling [134]. In contrast, additional knockdown of Oma1, maintaining the pro-fusion function of OPA1 restores normal mitochondrial morphology and cardiac function, suggesting that inhibition of mitochondrial fission is helpful to alleviate the symptoms of heart failure [134]. Furthermore, mitochondria-targeted administration of coenzyme Q10 (CoQ10) can also help alleviate the symptoms of heart failure by relieving oxidative stress or by regulating MCU to maintain [Ca^2+^]_m_ homeostasis [135,136], suggesting that regulating mitochondrial metabolism may be a promising therapeutic strategy for heart failure.

## 5. Mitochondria-Targeted Therapeutic Agents and Strategies in CVD

Now that, we have shown a large body of evidence that mitochondrial metabolism dysfunction has a major role in the pathology of CVD, we are going to discuss the mitochondrial therapeutic agents and strategies. Current strategies for mitochondrially targeted therapy mainly focus on factors of mitochondrial damage, such as decreased ATP/ADP ratio, signal pathway changes caused by insufficient NAD^+^, ROS-induced metabolic disorders, and [Ca^2+^]_m_ disorders caused by abnormal calcium channels which lead to strategies for mitochondria-targeted drug delivery: (1) cationic carriers designed using mitochondrial membrane potential coupled with drugs, such as mitoquinone (MitoQ) coupled with lipophilic triphenylphosphonium (TPP^+^), (2) mitochondrially targeted signal peptides as carriers for drug delivery, such as Szeto-Schiller peptide 31 (SS-31), (3) liposomes as carriers, such as mitochondria-targeted dendritic lipopeptide liposomal delivery platform [137], (4) nanoparticles coated with drugs for delivery, such as mitochondria-targeted nanoparticles (CsA@PLGA-PEG-SS31) [138], (5) physical invasion methods, such as microinjection of mitochondria directly into cells [139], (6) gene therapy technology, such as mitoTALENs technology [140]. Over the past two decades, scientists have made efforts to explore drugs that target cardiovascular mitochondria, and it is exciting to note that Imeglimin, a first-in-class mitochondria-targeted reagent for type 2 diabetes developed by Poxel SA, a French biopharmaceutical company, has been approved for marketing in Japan in 2021 and has shown favorable safety and tolerability. Other mitochondria-targeted therapies are also under intensive development [141], and some of which have already entered clinical trials. We believe that more high-quality mitochondria-targeted drugs will be available for the treatment of CVD in the future.

### 5.1. Potential Mitochondrial Targets for CVD

As mentioned above, the regulation of mitochondrial metabolism relies on mtDNA stability, [Ca^2+^]_m_, mitochondrial dynamics, mtROS, mitophagy, etc. These factors cooperatively maintain mitochondrial homeostasis. In addition, these factors are also widely involved in the progression of CVD, and therefore these factors are potential targets for CVD therapy. Here we briefly describe the role of these potential targets in regulating mitochondrial metabolism and the treatment of CVD.

#### 5.1.1. MPTP

MPTP, a complex assembled by a group of proteins, is a non-specific channel located between the inner and outer membranes of mitochondria. Numerous studies have shown that MPTP plays important roles in mitochondria-induced apoptosis [142,143]. Under stress conditions, mitochondrial oxidative stress can lead to [Ca^2+^]_m_ overload by increasing [Ca^2+^]_m_ uptake, causing the opening of MPTP, which in turn leads to a decrease in mitochondrial membrane potential and release of cytochrome C, triggering of mitochondrial apoptosis signal to induce cell apoptosis [144,145]. Likewise, MPTP contributes to cardiac I/R injury [146], atherosclerosis [147] and PAH [148] by causing mitochondrial depolarization and loss of mitochondrial function. Therefore, targeting MPTP is a rational strategy for the treatment of CVD. Cyclosporin A (CsA), a well-known inhibitor of MPTP, is widely used in the treatment of CVD [149], especially in acute myocardial injury [150]. However, unwanted inhibitory effect of CsA on the body’s immune system limits its application. Therefore, scientists design mitochondria-targeted CsA nanoparticles conjugated with poly-lactic/glycolic acid (PLGA) nanoparticle or SS-31, and the in vivo results show a good cardioprotective effects [138,151].

#### 5.1.2. Sirtuin 3 (Sirt3)

Acetylation of mitochondrial proteins is involved in the pathogenesis of CVD [152]. Sirt3 is a member of the histone deacetylase sirtuin family localized in mitochondria. Proteomic analysis reveals that most of the mitochondrial proteins can be modified by acetylation, and Sirt3 plays an important role in the deacetylation of these proteins [153]. It has been found that the protein expression of Sirt3 is decreased in failing hearts, accompanied by an increase in global protein acetylation level. Sirt3 knockdown increase global mitochondrial protein acetylation level, including metabolism-related key enzymes PDH and ATP synthase whose acetylation modification reduces the enzymatic activities. In addition, Sirt3 can enhance the expression of OXPHOS-related proteins by promoting mtDNA transcription via deacetylation of leucine-rich protein 130 and peroxisome proliferator-activated receptor gamma coactivator 1-alpha, and promote ATP generation by acting on complex I [154]. These results indicate that the deacetylation of mitochondrial proteins by Sirt3 is involved in mitochondrial ATP synthesis. Besides, studies have shown that the dysfunction of Sirt3 is associated with the progression of heart failure, hypertension and other CVD [155,156], and enhancing Sirt3 function could alleviate the symptoms of CVD [156]. These studies suggest that Sirt3 is a potential target for CVD therapy [155].

#### 5.1.3. [Ca^2+^]_m_

[Ca^2+^]_m_ homeostasis is vital for mitochondrial metabolism and CVD [56,57]. In recent years, methods of restoring [Ca^2+^]_m_ imbalance, namely modulating mitochondrial calcium uptakes/release channels such as VDACs, MCU or NLX, have attracted a lot of attentions. It is considered that [Ca^2+^]_m_ overload caused by MCU complex activation is the underlying mechanism of cardiac I/R injury. Su Li et al. find that in the cardiac microvascular ECs, overexpression of histidine triad nucleotide-binding 2 can directly interact with MCU and inhibit [Ca^2+^]_m_ overload caused by MCU activation, and in turn, it inhibits [Ca^2+^]_m_ overload-induced mitochondrial fission and apoptosis and attenuates I/R induced cardiac microvascular injury [157]. Similarly, overexpression of Parkin inhibits the increase of VDAC1 expression and [Ca^2+^]_m_ overload, and improves cardiac systolic dysfunction [158], while NCX inhibitor alleviates cardiac I/R injury to some extent [159]. So, methods to restore [Ca^2+^]_m_ homeostasis may also be a potential therapeutic strategy for CVD.

#### 5.1.4. Mitochondrial Dynamics

Mitochondrial dynamics is mainly characterized by a dynamic balance between mitochondrial fission and fusion. Mitochondrial fission and fusion have an important role in the mitochondrial material exchange, repair, and removal of damaged mitochondria, and are an important means for the body to maintain mitochondrial health. Extreme states of excessive fission or fusion are detrimental to health [160]. Under physiological or pathological conditions, mitochondrial fission and fusion form new homeostasis to meet the energy or physiological needs of the cell. For example, studies show that under pathological conditions or in case of rapid cell proliferation, mitochondria tend to increase division and decrease fusion levels [161,162]. In recent years, many studies have also shown that the level of mitochondrial fission is significantly increased in the development of CVD [160], and inhibiting mitochondrial fission or promoting mitochondrial fusion is beneficial to alleviate the symptoms of CVD [26,163]. Therefore, regulating mitochondrial fission and fusion status may be a potential target for the treatment of CVD.

### 5.2. Mitochondria-Targeted Agents for CVD

#### 5.2.1. CoQ10 and MitoQ

CoQ10 is one of the electron carriers in the mitochondrial ETC. It is involved in mediating the electron transfer between mitochondrial OXPHOS complex I/II and complex III and plays an important role in maintaining mitochondrial aerobic respiration and ATP production. CoQ10 is widely used in the treatment of metabolism-related diseases because of its powerful antioxidant protective effect, which protects mitochondria from oxidative damage to maintain normal mitochondrial metabolic function. The protecting role of CoQ10 in CVD has been well reviewed recently [164]. Due to the double membrane structure of mitochondria, CoQ10 cannot effectively enter mitochondria, resulting in low bioavailability. Scientists synthesize MitoQ with mitochondria-targeted function by covalently coupling the benzoquinone part of CoQ10 and TPP^+^ through the deca-carbon aliphatic chain by taking advantage of the mitochondrial membrane potential targeting property of TPP^+^. MitoQ has been widely used in the treatment of various metabolic syndromes [165]. Evidence from preclinical and clinical trials has shown promising protective effects of MitoQ in CVD. In type II diabetic rats, MitoQ can alleviate cardiac I/R injury by enhancing pink1/Parkin-mediated mitophagy. Other reports also study that MitoQ by enhancing mitochondrial aerobic respiration and reducing ROS could improve the cardiac insufficiency induced by hypertension [166], cardiac fibrosis, and left ventricular dysfunction induced by pressure overload [167,168]. In the vascular system, it has been reported that MitoQ could inhibit platelet activation by reducing platelet adhesion and diffusion on collagen surface, reducing the expression of P-selectin and CD63, and inhibiting platelet aggregation induced by collagen [169]. Moreover, MitoQ can inhibit ROS, reduce vascular SMC apoptosis, inhibit vascular calcification [170], and enhance vascular compliance in elderly patients [171,172]. Furthermore, MitoQ alleviates ROS in acute hypoxic pulmonary vasoconstriction and right ventricular remodeling induced by chronic hypoxia. In a hypoxia-induced rat PAH model, MitoQ inhibits the metabolic conversion of pulmonary microvascular ECs in rats with pulmonary hypertension by removing ROS, reducing [Ca^2+^]_c_, inhibiting mitochondrial fission, and improving mitochondrial aerobic respiration [173]. MitoQ has also been widely tested in clinical trials for CVD (NCT04109820, NCT03586414, NCT03506633, NCT04851288, NCT05561556, NCT05410873, NCT02966665, NCT02690064). These studies fully demonstrate that MitoQ has broad prospects in the treatment of CVD in the future.

#### 5.2.2. Melatonin

Melatonin is a neuroendocrine hormone produced by the pineal gland of mammals and its physiological functions include neuroregulation, sleep promotion, antioxidant, anti-inflammation, anti-aging, immune regulation, endocrine regulation and cell growth promotion [174]. The level of melatonin varies greatly among different populations and decreases gradually with aging. Melatonin has amphipathic properties enabling itself penetrating through a variety of cell membranes. Melatonin can quickly cross the cellular and mitochondrial membranes and accumulate in mitochondria [175]. Due to its powerful antioxidant and anti-inflammatory properties, melatonin has been widely used in therapeutic studies of CVD. Encouragingly, most studies have found a protective effect of melatonin in heart failure [176], cardiac I/R injury [177], aneurysm [178,179], atherosclerosis [180], and PAH [181,182]. These preclinical in vivo studies of melatonin efficacy in CVD over the past 5 years are summarized in Table 1.

In addition, the therapeutic effects of melatonin on various CVD are being widely carried out in the clinic. The results of some clinical trials have also shown that melatonin has improvements in some CVD [201]. We have searched “ClinicalTrials.gov” for the keywords “melatonin” and “cardiovascular disease” and have found 54 clinical trial results. The information on clinical trials conducted over the past 5 years about the therapeutic effects of melatonin in CVD is summarized in Table 2.

#### 5.2.3. SS-31

SS-31 (elamipretide) is a cell-permeable aromatic cationic tetrapeptide. Under physiological conditions, SS-31 carries three positive charges, and can selectively target and accumulate in the inner mitochondrial membrane through electrostatic and hydrophobic interactions [202]. SS-31 contains dimethyltyrosine residues, which can interact with oxygen-free radicals in mitochondria to form inactive tyrosine radicals and exhibit powerful antioxidant capacity. SS-31 can specifically bind to cardiolipin, inhibit cardiolipin oxidation, reduce electron leakage, promote ATP generation, and maintain mitochondrial structure and function.

SS-31 is widely used in CVD treatment. In the aging heart, SS-31 reduces oxidative stress and mitochondrial electron leakage, inhibits the opening of MPTP, improves cardiac energy metabolism, restores mitochondrial activity, promotes ATP generation, and alleviates cardiac diastolic function [203,204,205]. Cardiac proteomics studies have found that SS-31 ameliorates aging-induced changes in the post-translational modification profile of cardiac proteins [206,207], improves transverse aortic constriction (TAC)-induced mitochondrial damage and heart failure phenotypes, and suppresses mitochondrial proteomic changes [208]. Preclinical studies have shown that SS-31 by its anti-inflammatory and anti-oxidant effects alleviates sepsis-induced cardiac injury [209], improves failing heart function, ameliorates myocardial mitochondrial fragmentation [210], maintains myocardial mitochondrial integrity, and improves myocardial energy supply in rats with I/R [211,212]. In addition, SS-31 can also remove ROS, stabilize mitochondrial membrane potential, reduce cardiomyocyte apoptosis, and alleviate diastolic cardiomyopathy and myocardial toxicity induced by doxorubicin in rats [213,214]. In high-fat diet ApoE knockout mice, administration of SS-31 inhibits cholesterol uptake, which in turn inhibits foam cell formation and atherosclerosis progression [215]. In TAC-induced PAH mice model, SS-31 can inhibit the release of inflammatory factors and improve endothelial function by reducing the expression of pro-oxidant protein NOX1/NOX2, and reduce TAC-induced right ventricular systolic pressure [216]. These results indicate that SS-31 has great potential in the treatment of CVD.

A clinical trial of SS-31 on the improvement of mitochondrial function in heart failure patients after transplantation finds that SS-31 treatment significantly improves myocardial mitochondrial oxygen flux, activity of complex I and complex IV, and mitochondrial energy supply [217]. In addition, SS-31 improves cardiac function in patients with myocardial infarction [218]. A phase 2a trial evaluating the safety, tolerability, and efficacy of SS-31 in ST-elevation myocardial infarction patients, SS-31 has been found to be relatively safe and well tolerated. However, SS-31 treated group hasn’t shown an improvement in pre-specified magnetic resonance imaging, angiographic, electrocardiographic, or clinical outcomes [219]. These results indicate that SS-31 is inconsistent with different therapeutic indicators for the same disease. A phase 2 trial of left ventricular dysfunction in heart failure patients with reduced ejection fraction, SS-31 has been well tolerated but shows no significant improvement in left ventricular end systolic volume and left ventricular ejection fraction compared with placebo group [220]. These results suggest that additional clinical trials are needed to fully evaluate the true therapeutic effects of SS-31 in different CVD.

#### 5.2.4. MitoTEMPOL

MitoTEMPOL is a novel mitochondria-targeted antioxidant formed by the combination of the superoxide dismutase mimics Tempol and TPP^+^. MitoTEMPOL can scavenge mtROS, so it can alleviate the oxidative damage of mitochondria induced by mtROS under various pathological conditions, restore mitochondrial function, and maintain normal function of the body.

In CVD, MitoTEMPOL could improve ventricular arrhythmias, eliminate sudden cardiac death, and inhibit the expression of proteome remodeling and specific phosphoproteome alterations induced by chronic heart failure in guinea pigs, or preventing and reversing heart failure by removing ROS in the heart. Mechanistically, MitoTEMPOL repairs the disruption of normal coupling between cytoplasmic signaling and nuclear genetic programs, and restores the mitochondrial function, antioxidant enzymes, Ca^2+^ handling, and action potential repolarization disrupted by mtROS [221]. By clearing mtROS, MitoTEMPOL can inhibit the opening of the MPTP in the rat heart induced by nicotine, inhibit cardiac hypertrophy and cardiac fibrosis, and improve the sensitivity of rat myocardium to I/R injury caused by nicotine [222,223]. In atherosclerosis, MitoTEMPOL improves the vulnerability characteristics of atherosclerotic plaques in ApoE^-/-^/SOD2^+/-^ mice by reducing the expression of calpain-2, caspase-3 and matrix metalloproteinase-2, smooth muscle cell apoptosis and matrix degradation [224]. By removing mtROS, MitoTEMPOL reverses the senescence of vascular SMCs caused by ALDH2 deficiency and enhances plaque stability [225]. MitoTEMPOL reduces the ability of LPS-activated monocytes in inducing inflammation in ECs and thus maintains the normal function of vascular ECs. In addition, MitoTEMPOL can reverse the inhibition of autophagic flux in macrophages by ox-LDL through mTOR signaling pathway and promote autophagy-mediated lipid degradation in macrophages. MitoTEMPOL also increases cholesterol efflux by upregulating autophagy-dependent ABCA1 and ABCG1 to prevent macrophages from transforming into foam cells [226]. All these results indicate that MitoTEMPOL has valid cardiovascular protective effect.

#### 5.2.5. MitoSNO

Nitrosylation is one of the common post-translational modifications of proteins, which can regulate enzyme activity, subcellular localization and protein interactions. Studies have shown that nitrosylation modifications are also widely present in mitochondrial proteins and participate in the regulation of mitochondrial metabolic enzyme activities [227]. The reversible S-nitrosation modification is the main way that nitric oxide (NO) metabolism regulates mitochondrial physiology and pathology [227,228]. To further investigate the role of mitochondrial S-nitrosation, mitochondria-targeted MitoSNO has been formed by combining the NO donor SNAPNO (S-nitroso-N-acetylpenicillamine) and lipophilic TPP^+^. Driven by the positive charge of TPP^+^ and the mitochondrial inner membrane potential, MitoSNO accumulates rapidly in the mitochondria. Further study finds 13 S-nitrosation mitochondrial proteins, including aconitase, mitochondrial dehydrogenase and α-ketoglutarate dehydrogenase, whose activities are selectively and reversibly inhibited by S-nitrosation, indicating that S-nitrosation is reversibly involved in the regulation of enzyme activity of mitochondrial core metabolic enzymes [229].

In cardiac I/R injury, a variety of NO donors and S-nitrosation agents can protect ischemic myocardium from infarction [230,231]. MitoSNO pretreatment 5 min before I/R could significantly improve cardiac I/R injury [229]. The protective effect of MitoSNO is due to the reversible S-nitrosation modification of mitochondrial complex I. MitoSNO inhibits the activity of complex I ND3 subunit through Cys39 reversible S-nitrosation modification, which slows mitochondrial reactivation during the critical first minutes of ischemic tissue reperfusion, thereby reducing ROS production, oxidative damage, and tissue necrosis [232]. Further studies find that mitochondria-targeted S-nitrosation can prevent heart failure and myocardial fibrosis after long-term myocardial infarction [233]. These results suggest that targeting S-nitrosation in mitochondria may be a new strategy for the treatment of cardiac I/R injury.

### 5.3. Mitochondria-Targeted Gene Therapy Strategies

As mentioned above, we have introduced some therapeutic drugs targeting mitochondria for CVD and most of them are antioxidants and the accurate mechanism for cardiovascular protection is unclear, so there may be risks in their use. Theoretically, mitochondria-targeted gene therapy can target specific abnormal mitochondrial genes for repair and thus achieve a true “targeted therapy”. Thanks to the development of new gene sequencing technology, the identification of mtDNA pathological variations have become routine and easy. However, due to the heterogeneity of the mitochondrial genome, the precautionary strategy of mtDNA-mediated mitochondrial diseases is still suboptimal. Therefore, in this part, we overview the current technical means of mitochondrially targeted gene therapy and their limitations of the application.

#### 5.3.1. Mitochondria-Targeted Gene Editing Technologies

Because of the heterogeneity of mtDNA, mtDNA mutations need to accumulate to a threshold before they cause mitochondria-related diseases. For example, proband’s fibroblasts pathogenic variant m.8993T > G variant in MT-ATP6 subunit is 83%, and exhibits severe defects in mitochondrial cristae structure and decreased spare respiratory capacity in response to energy stress [234]. Therefore, it is a good idea to repair or excise mutated mtDNA by gene-editing technology to reduce the proportion of mutated mtDNA in the overall mtDNA. By combining an adeno-associated virus (AAV) 9-loaded TALEN (mitochondria-targeted TALEN, MitoTALEN) gene-editing tool with a mitochondrial guiding sequence, the m.5024C > T mutation site is successfully targeted to be cleaved, thus achieves the purpose of inhibiting the replication of the mutant gene. Six months after injection of this gene editing tool into the m.5024C > T mutant mouse model, the mtDNA mutation rate in mouse skeletal and heart muscle tissue decreases by 50%, most of the mutant genes are eliminated, and the mutation-induced reduction in transfer RNA^Ala^ levels is ameliorated [235]. Similarly, using AAV9.45 to introduce zinc finger nuclease (mitochondrially targeted zinc finger nuclease, MitoZFN) successfully reduces the mtDNA mutation rate in cardiac tissues by approximately 40%, results in phenotype rescue, that is characterized by an increase of pyruvate and a decrease of lactate levels, suggesting an improvement of mitochondrial respiration [236]. This targeted degradation of defective mtDNA using nuclease technology significantly reduces the level of defective mtDNA in cells and alleviates the disease caused by mtDNA defects. However, such nucleases have specific DNA recognition sequences and only recognize and repair specific mutations. In addition, cleavage degradation dose not truly repair the mutated DNA, and may lead to heterogeneous shift mutations, which limits the use of these techniques to some extent.

CRISPR gene editing technology has been widely confirmed in nuclear DNA editing, which has the advantages of easy design, simple operation and efficient gene editing. Using CRISPR technology on mtDNA damage repair has been conducted. For example, Yunjong Lee et al., successfully locate regular FLAG-Cas9 to mitochondria to edit the line mtDNA of Cox1 and Cox3 genes using sgRNAs targeting the mitochondrial genome, leading to mtDNA breaks at specific sites, resulting in disruption of mitochondrial protein homeostasis [237]. A mitochondria-targeted Cas9 (mitoCas9)-induced reduction of mtDNA disrupts mitochondrial membrane potential and inhibits cell growth [237]. However, the highly negatively charged sgRNA is difficult to penetrate the mitochondrial bilayer structure to enter the mitochondrial matrix, which limits the application of CRISPR gene editing technology in mtDNA to some extent. Given this, Mok et al. develop an efficient CRISPR-free gene editing tool, “The all-protein cytosine base editor DdCBE”, and achieve precise editing effect on multiple mitochondrial genes, which affects oxygen consumption rate [238,239]. This provides a new technology for future mtDNA precision editing and repair.

#### 5.3.2. Ectopic Expression of Mitochondrial Proteins

Ectopic expression of mitochondrial proteins is the process of constructing recombinant plasmids by combining mtDNA sequences encoding mitochondrial proteins and mitochondria-targeted sequences and integrating them into genomes through vectors (usually AAV) so that proteins originally expressed in mitochondria are sent to mitochondria through mitochondria-targeted peptides after being expressed in the nucleus and cytoplasm of cells to ameliorate metabolism dysfunction caused by impaired mitochondrial self-expression. Giovanni Manfredi et al. use ectopic expression of wild-type mtDNA encoding ATPase 6 of respiratory chain complex V in the nucleus and successfully import them into mitochondria, which significantly improve OXPHOS and ATP production in cells containing the ATP6 DNA 8993T > G mutation [240]. This technique allows the personalized design of recombinant plasmids to repair mitochondrial metabolic dysfunction caused by defects in mtDNA and is a potential mitochondria-targeted gene therapy strategy. However, this technique has some shortcomings, such as the ectopically expressed mitochondrial protein may not assemble correctly as expected when binding to the mitochondria-targeted peptide, and the control of protein expression by the ectopically expressed mitochondrial protein is also a factor should be considered, because mitochondrial metabolism is a strictly fine regulated process, any increase or decrease in any component or protein involved in OXPHOS may result in an excess or deficiency of other intermediate metabolites, ultimately reducing metabolic efficiency. Therefore, the influence of these potential factors should be fully considered in the design of these technologies.

#### 5.3.3. Mitochondrial Replacement Therapy (MRT) Technologies

The limitations of the conventional gene therapy approach make them fall short of expectations in the treatment of pathogenic mtDNA-defected diseases. MRT is the most effective technique for the prevention of hereditary mtDNA mutation disorders. This technique has been successfully tested in rhesus macaque monkeys [241] by transferring the nuclear genome of a mother carrying a pathogenic mtDNA mutation into the enucleated oocytes of another healthy mother by microinjection. However, the “three-parent” concept raised by this technique is controversial in terms of reproductive ethics. In addition, due to the possibility of selective replication and genetic drift, small amounts of residual maternally defective mtDNA may still gradually reach to pathogenic levels. Therefore, XiaoYan Fan et al. inject transmembrane peptide mRNA fused with autophagy receptor during nuclear transfer to reduce the carrying of defective mtDNA in reconstructed embryos by forcibly inducing the degradation of donor mitochondria [242]. To avoid ethical problems, scientists have developed co-culture technology to allow mtDNA-deficient cells to obtain autologous healthy mitochondria [243], or cytosolic fusion technology to fuse normal cells with mtDNA-deficient cells to form healthy hybrid cells to compensate for mitochondrial dysfunction in mtDNA-deficient cells [244]. In a New Zealand white rabbit model of I/R injury, the transplanted autologous isolated healthy mitochondria into the I/R area mainly exist in the interstitial space and have been internalized by cardiomyocytes within 2–8 h, which successfully enhance myocardial oxygen consumption and ATP production, and improve myocardial function after I/R injury [245]. Microinjection has also been used to directly transfer healthy mitochondria into a fertilized egg, but it has not been widely used because it can only transfer single mitochondrion into a cell, which is inefficient and prone to damage to the recipient cells. For this reason, Wu et al. introduce mitochondria with desired mtDNA haploids into recipient cells by photothermal nanoblade technology under the action of laser pulses and successfully rescue the pyrimidine auxotroph phenotype and respiration in cells that lack mtDNA [246]. In CVD, mitochondrial transfer technology has multiple protective effects, mainly include: (1) improving the mitochondrial biogenesis, (2) enhancing the antioxidant capacity and (3) reducing the cell apoptosis [247], which ultimately rescue the mitochondrial metabolism and cardiovascular function [248,249]. The mitochondria-targeted therapeutic strategies in CVD are illustrated in Figure 2. These new technologies undoubtedly provide rich technical means for mitochondria-targeted gene therapy, but most of these technologies are still in the preclinical research stage. The mitochondria introduction mechanism and potential toxic side effects of some technologies are still unclear and need to be tested in more practice.

## 6. Prospection

Overall, mitochondrial metabolism is essential for the maintenance of normal cardiovascular function. A great deal of effort has been invested in elucidating the mechanism of mitochondrial metabolism involved CVD and developing novel mitochondria-targeted drugs and technologies over the past decades. Compared with traditional symptomatic treatments with mitochondria-targeted delivery drugs, mitochondria-targeted gene therapy technology requires clarifying the mechanism of the disease and treating the cause at the genetic level, which is a major advancement of science and technology in the treatment of CVD. The fact to note is that most studies are still at the preclinical stage, some mitochondria-targeted drugs, such as mitochondria-targeted antioxidants, that has cardiovascular protection are only symptomatic rather than pathogenetic treatment in the true sense. In contrast, gene therapy theory remains to be ultimate strategy for all-cause treatment, but to date, mitochondria-targeted gene therapy trials mainly focus on treating diseases that affect specific tissues, such as eye diseases and neurological-related diseases, for which genes are simply delivered to very limited locations. However, CVD which often involves multiple organ systems, usually requires systemic targeting of therapy. On one hand, systemic treatment greatly increases the number of therapeutic genes used and the cost of treatment. On the other hand, the tissue offtargeting and immune response risks of systemic gene therapy also imposes great challenges. Therefore, mitochondria-targeted gene therapy for CVD still has a long way to go.

## Figures and Tables

**Figure 1 pharmaceutics-14-02760-f001:**
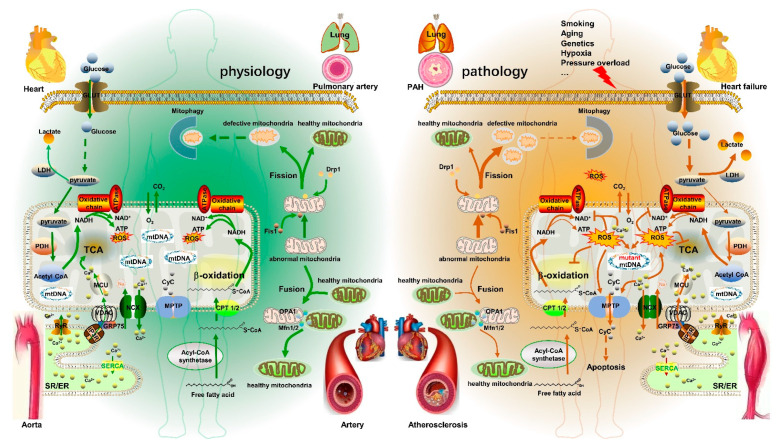
The major factors and pathways involved in mitochondrial metabolism in the physiological and pathological conditions of the cardiovascular system. Under physiological conditions, most cells obtain energy through glucose OXPHOS in mitochondria. Under pathological conditions, pathological stimuli (such as smoking, aging, et al.) disrupt intracellular Ca^2+^ homeostasis. Once the [Ca^2+^]_c_ concentration exceeds a certain threshold, it could cause [Ca^2+^]_m_ overload which inhibits the activities of mitochondrial metabolism-related calcium-sensitive enzymes, resulting in the reduction of OXPHOS efficiency and ATP production, and promoting the generation of ROS. Excessive ROS: (1) inhibit the activities of the mitochondrial metabolism-related enzymes and reduce the mitochondrial metabolic activity, (2) cause oxidative damage and degradation of mtDNA, promote mitochondrial division, and ultimately cause mitochondrial dysfunction, (3) promote MPTP opening, leading to cytochrome C leakage and inducing apoptosis. In CVD, mitochondrial dysfunction causes metabolic reprogramming, which contributes to the development of CVD. GLUT: glucose transporter; LDH: lactic dehydrogenase; PDH: pyruvate dehydrogenase; TCA: tricarboxylic acid cycle; RyR: ryanodine receptor; CyC: cytochrome C.

**Figure 2 pharmaceutics-14-02760-f002:**
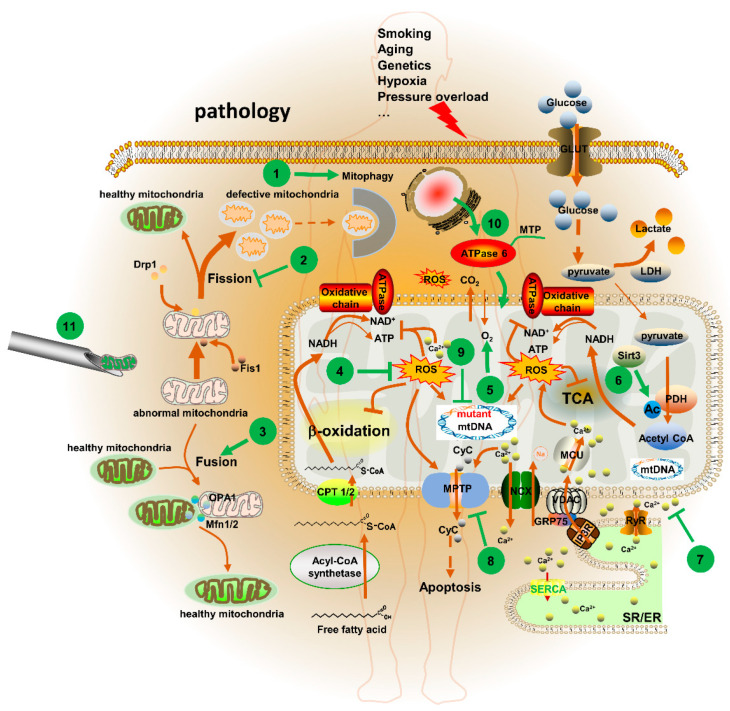
Mitochondria-targeted therapeutic strategies in CVD. 1. Enhancing mitophagy to clear defective mitochondria. 2. Inhibiting mitochondrial fission to alleviate the fragmentation of mitochondria. 3. Promoting mitochondrial fusion to repair damaged mitochondria. 4. Clearing ROS with mitochondria-targeted ROS scavenger, such as MitoQ, melatonin, SS-31 and MitoTEMPOL, to protect mitochondrial metabolism-related proteins, enzymes and mtDNA from oxidative damage. 5. Restoring O_2_ supply to enhance OXPHOS and reduce glycolysis. 6. Correcting posttranslational modification with Sirt3 agonist or MitoSNO to restore activities of mitochondrial metabolism-related proteins and enzymes. 7. Maintaining the homeostasis of Ca^2+^ to avoid [Ca^2+^]_m_ overload. 8. Inhibiting MPTP opening with mitochondria-targeted CsA nanoparticles to avoid cell apoptosis. 9. Deleting the mutant mtDNA with mitochondria-targeted gene editing technologies, such as MitoTALEN and MitoZFN, to reduce the ratio of mutant mtDNA. 10. Extopic expression of mitochondrial proteins to restore the function of mitochondria. 11. Using mitochondrial replacement technologies to cure hereditary mtDNA mutation disorders.

**Table 1 pharmaceutics-14-02760-t001:** The latest preclinical efficacy of melatonin in CVD.

Disease	Animals Model	Results and/or Possible Mechanism	Reference
Heart diseases	Doxorubicin induced cardiotoxicity in rats	Melatonin exerts cardioprotective efficacy by preserving mitochondrial function and dynamics	[183,184]
	Cardiac I/R injury in prediabetic obese rats	Melatonin exerts cardioprotective effects against cardiac I/R injury in prediabetic obese rats by activation of modulating melatonin receptor 2	[185]
	Myocardial I/R injury in a DCD heart rat model	Melatonin exerts cardioprotective effects by activation of the JAK2/STAT3 signaling pathway	[186]
	MI rat model induced by ligation of the left anterior descending coronary artery	Melatonin exerts cardioprotective effects by promoting cardiomyocyte proliferation and heart regeneration, and by inhibiting pyroptosis and cardiac fibrosis	[187,188,189]
	A murine cardiac hypertrophy model by transverse aortic constriction	Melatonin ameliorates pressure overload-induced cardiac hypertrophy by attenuating Atg5-dependent autophagy and activating the Akt/mTOR pathway	[190]
	Isoprenaline hydrochloride induced acute heart failure in aged rat	Melatonin prevents the mitochondrial dysfunction by increasing the CNPase level	[191]
Atherosclerosis	High-fat diet treated ApoE knockout mice	Melatonin decreases pro-inflammatory macrophage polarization, vulnerable plaque rupture, endothelial cell injury and atherosclerotic plaque	[192,193,194,195,196]
Aneurysm	Rat AAA model	Melatonin prevents AAA formation through anti-inflammatory, antioxidative and anti-autophagy effects	[179,197]
	TAAD mouse model	Melatonin exerts therapeutic effects against TAAD by reducing oxidative stress and SMC loss via activation of Sirt1 signaling	[178]
PAH	Hypoxia-induced PAH in mice	Melatonin inhibits inflammasome-associated activation of endothelium and macrophages in PAH	[181]
	Dasatinib aggravated rat experimental model of hypoxic PAH	Melatonin attenuates dasatinib-induced PAH by inhibiting oxidative damage and apoptosis of ECs and inhibiting abnormal proliferation of SMCs	[198]
	MCT-induced rat PAH model	Melatonin treatment reduces MCT-induced RV hypertrophy, fibrosis, and remodeling	[199]
	A chronically hypoxia-induced PAH of the newborn sheep model	Melatonin improves vasodilatory function by enhancing the vasodilator prostanoid pathway	[200]

I/R: ischemia/reperfusion, TAAD: thoracic aortic aneurysm and dissection, AAA: abdominal aortic aneurysm, MI: myocardial infarction, MCT: monocrotaline.

**Table 2 pharmaceutics-14-02760-t002:** The latest clinical trials of melatonin in CVD.

Disease	Study Title	Melatonin Dose	Outcome	Clinical Trials.gov Identifier
Heart failure	Effect of melatonin on cardiovascular and muscle mass and function in patients with heart failure	10 mg tablets orally every night for 24 weeks	Unknown	NCT03894683
Coronary artery disease	Melatonin impact on the outcomes of myocardial I/R injury during coronary artery bypass grafting surgery	60 mg/day 5 days before surgery	Ongoing	NCT05552586
Coronary artery disease	Enhanced recovery after surgery in coronary artery bypass graft/off-pump coronary artery bypass	5 mg in the evening	Unknown	NCT03956420
Coronary artery disease	Melatonin on coronary artery calcification	3 mg/day for 6 months	Unknown	NCT03966235
Acute coronary syndrome	Effects of melatonin on reperfusion injury	Intravenous 11.61 mg	Unknown	NCT03303378
Essential hypertension	Melatonin and essential arterial hypertension	1 mg/day for 1 year	Ongoing	NCT05257291
Hypertension	Trial of oral melatonin in elevated blood pressure	3 mg for three weeks	Ongoing	NCT03764020

## Data Availability

Not applicable.

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
