# Peer review of "The Therapeutic Strategies Targeting Mitochondrial Metabolism in Cardiovascular Disease"

_pharmaceutics, 2022, doi:10.3390/pharmaceutics14122760_

Round 1

Reviewer 1 Report

The paper is sound in providing therapeutic strategies to CVD. It requires illustrations & figures to well explain the mechanism of mitochondrial pathway. Review article is incomplete without proper figures. 

References are good in number but where therapeutic strategies are mentioned add recent references related to clinical trials.

Mention the impact and significance of novel therapeutic strategies compared to older techniques.

Reviewer 2 Report

The authors in the manuscript review the role of mitochondrial metabolism in the prevention and treatment of CVD, with a specific overview for the mitochondria-targeted therapeutic strategies.
In the first part of manuscript the authors describe the causes of CVD and the possible role of mitochndrial disfunctions that underpinning the cardiovascular diseases.
The second part the ptential mitochondrial target for theraputic treatements of CVD, most of them are antioxidants are listed In the third part of review the authors describe strategies for mitochondria-targeted gene therapies.
In particular authors overview the current technical for mitochondrial gene therapies and their limitations.

The review is well organized and it should be of great interest for the readers.

Minor issues

page 4, second paragraph: word "pyruvate" is reported two times.
page 6, second paragraph: CPT1 is not a transporter, but an enzyme that catalyzes the transfer of acyl group from acyl-CoA to carnitine.
page 10 "5.1.3[. Ca2+]", should be "5.1.3.[Ca2+]"
page 10, last paragraph, CoQ10 is not an enzyme, but it is an essential cofactor in the mitochondrial electron transport chain.
page 14, The authors report that the gene therapies have effects on the rescue of mtDNA mutations.
The authors should report also, if the gene therapies rescue the cardiovascular phenotypes.
